# RNAseq Analysis of Novel 1,3,4-Oxadiazole Chalcogen Analogues Reveals Anti-Tubulin Properties on Cancer Cell Lines

**DOI:** 10.3390/ijms241411263

**Published:** 2023-07-09

**Authors:** Stefano Zoroddu, Luca Sanna, Valentina Bordoni, Weidong Lyu, Gabriele Murineddu, Gerard A. Pinna, Sonia Vanina Forcales, Arturo Sala, David J. Kelvin, Luigi Bagella

**Affiliations:** 1Department of Biomedical Sciences, University of Sassari, Viale San Pietro 43/b, 07100 Sassari, Italy; s.zoroddu3@studenti.uniss.it (S.Z.); lusanna@uniss.it (L.S.); v.bordoni@studenti.uniss.it (V.B.); 2Division of Immunology, International Institute of Infection and Immunity, Shantou University Medical College, Shantou 515031, China; lvweidong1230@126.com (W.L.); dkelvin@jidc.org (D.J.K.); 3Department of Medicine, Surgery and Pharmacy, University of Sassari, 07100 Sassari, Italy; muri@uniss.it (G.M.); pinger@uniss.it (G.A.P.); 4Department of Pathology and Experimental Therapeutics, School of Medicine, Health Science Campus of Bellvitge, University of Barcelona, Carrer de la Feixa Llarga, s/n, Hospitalet de Llobregat, 08907 Barcelona, Spain; sforcales@ub.edu; 5Centre for Inflammation Research and Translational Medicine (CIRTM), Department of Life Sciences, Brunel University, London UB8 3PH, UK; arturo.sala@brunel.ac.uk; 6Department of Microbiology and Immunology, Dalhousie University, 6299 South St, Halifax, NS B3H 4R2, Canada; 7Sbarro Institute for Cancer Research and Molecular Medicine, Centre for Biotechnology, College of Science and Technology, Temple University, Philadelphia, PA 19122, USA

**Keywords:** oxadiazole, cancer, microtubule, chemotherapy, RNAseq

## Abstract

1,3,4-Oxadiazole derivatives are among the most studied anticancer drugs. Previous studies have analyzed the action of different 1,3,4-oxadiazole derivatives and their effects on cancer cells. This study investigated the characterization of two new compounds named **6** and **14** on HeLa and PC-3 cancer cell lines. Based on the previously obtained IC_50_, cell cycle effects were monitored by flow cytometry. RNA sequencing (RNAseq) was performed to identify differentially expressed genes, followed by functional annotation using gene ontology (GO), KEGG signaling pathway enrichment, and protein–protein interaction (PPI) network analyses. The tubulin polymerization assay was used to analyze the interaction of both compounds with tubulin. The results showed that **6** and **14** strongly inhibited the proliferation of cancer cells by arresting them in the G2/M phase of the cell cycle. Transcriptome analysis showed that exposure of HeLa and PC-3 cells to the compounds caused a marked reprograming of gene expression. Functional enrichment analysis indicated that differentially expressed genes were significantly enriched throughout the cell cycle and cancer-related biological processes. Furthermore, PPI network, hub gene, and CMap analyses revealed that compounds **14** and **6** shared target genes with established microtubule inhibitors, indicating points of similarity between the two molecules and microtubule inhibitors in terms of the mechanism of action. They were also able to influence the polymerization process of tubulin, suggesting the potential of these new compounds to be used as efficient chemotherapeutic agents.

## 1. Introduction

In eukaryotic cells, microtubules, actin, and intermediate filaments play a pivotal role in an array of biological processes. The tubulin gene family appeared early during evolution [1,2] since it is fundamental for the architecture of cells and central for cell division and other biological features [3]. Thus, tubulins are extremely attractive as therapeutic targets in cancer [4]. Having a fundamental role in mitosis, microtubules are dynamic machines able to organize themself in the formation of the mitotic spindle. Furthermore, microtubules are involved in the transport of vesicles and movement of organelles, contributing to organism and cell homeostasis [3,5]. Microtubules have a tubular structure made up of polymers of *αβ*-tubulin heterodimers, which form 13 protofilaments [6]. These characteristics make microtubules important targets for cancer therapy [7] leading to the development of drugs known as microtubule targeting agents (MTAs). MTAs can display an efficient killing activity in both solid and hematological cancers [6,8,9]. Among the most famous MTAs are the taxane family, vinca alkaloids, and colchicine. They can be divided into microtubule-stabilizing agents (MSAs) and microtubule-destabilizing agents (MDAs). One of the main features of cancer cells is a hyperactive cell cycle activity, which induces cells to proliferate in an uncontrolled way [10], and tubulin plays a critical role in controlling mitosis. Consequently, tubulin has become a key target for drug development in cancer research [6,8,11,12]. Using a combination of proteomics, transcriptomics, biostatistical analysis, and computational chemistry, we and others are trying to understand how newly developed molecules can affect the behavior of cancer cells for therapeutic purposes [13]. In previous studies, we explored the mechanism of action and molecular targets of 1,3,4-oxadiazoles which showed antiproliferative activity in different types of cancer cells [7]. We identified a molecule called **2j**, which has been used as a lead compound in further studies [14]. Furthermore, our recent research identified an array of compounds in which the 1,3,4-oxadiazole ring has been changed to 1,3,4-thiadiazole (**6**) and 1,3,4-selenadiazole (Table 1). Our recent findings uncovered the antiproliferative effect of these two derivates, of which IC_50_s were calculated on HeLa and PC-3 cancer cell lines (Table 2) [1]. In the current study, the anticancer activity of these two compounds was investigated using biological and bioinformatic approaches.

## 2. Results

### 2.1. Cell Cycle Analysis

To understand the effect of the compounds **6** and **14** on the cell cycle, we used flow cytometry. In order to highlight effects on the cell cycle, we used a concentration of 1 µM for this set of experiments. The results showed that both compounds induced a strong perturbation of the cell cycle, with cells almost totally arrested in the G2/M phase. We observed the action of the compounds in a time course of 6, 12, and 18 h. Compound **14** was the strongest drug in both HeLa and PC-3 cells. Both compounds showed an initial effect on the cell cycle as early as 6 h with a gradual increase in G2/M phase. On HeLa cells after 18 h, both compounds showed a clear block with almost all cells in G2/M, while, on PC-3 cells, the effect was evident but less pronounced. We may conclude that the tested compounds mainly inhibit cancer cell proliferation by causing a mitotic block (Figure 1).

### 2.2. Analysis of Gene Expression after Treatment of HeLa and PC-3 Cells with Compounds ***6*** and ***14***

To study the effect of thiadiazole derivatives on gene expression we focused our attention on compounds **14** and **6** since they showed the strongest mitotic block effect in both cell lines. Cells were treated with a concentration of 1 μM of the two compounds at different time points (6, 12, and 18 h) and subjected to RNA sequencing. Gene expression was calculated using RSEM and represented by FPKM. Differential expression analysis was performed using edgeR. There was a total of 204 differentially expressed genes (DEGs) after treatment of HeLa cells with compound **14** for 6 h and a total of 273 DEGs after treatment for 18 h. A total of 258 DEGs were detected after treatment of PC-3 cells with compound **14** for 6 h and 279 DEGs after 12 h. Treatment of HeLa cells with compound **6** for 6 h caused 282 DEGs and, after 18 h, 331 DEGs were detected (Appendix A). A total of 241 DEGs were detected after 6 h of treatment of PC-3 cells with compound **6** and 122 after 12 h (Appendix A).

### 2.3. GO Enrichment and KEGG Pathway Analysis of DEGs

In order to understand the functions and further characterization of the DEGs, GO and KEGG analysis were carried out using online tools (OmicShare tools). GO enrichment analysis indicated that the biological processes regulated by compound **14** on both cell lines were mainly related to the mitotic cell cycle, programed cell death, and apoptotic processes. Molecular function analysis suggested that the most significant changes were at the level of microtubule motor activity and cyclin-dependent protein serine/threonine kinase activity. Accordingly, the most prominent cellular component analysis hits were microtubule, intracellular components, and chromosome (Figure 2A). KEGG pathway analysis showed that the differential genes were mainly involved in the following biological processes: regulation of mitotic cell cycle phase transition, DNA damage checkpoint, microtubule-based process, and positive regulation of cell death. The main signal pathways involved were cell cycle, FoxO signaling pathway, apoptosis, and p53 signaling pathway. Similar results were noted for compound **6**. Enrichment analysis of GO for biological processes was mainly related to the mitotic cell cycle, programed cell death, and apoptotic processes. Regarding the molecular function analysis, the most significant annotation was binding, microtubule motor activity, and cyclin-dependent protein serine/threonine kinase regulator activity. The most prominent annotations in cellular components analysis were microtubule, intracellular components, and chromosome (Figure 2B). KEGG pathway analysis showed that the DEGs were mainly involved in mitotic cell cycle phase transition, regulation of cell cycle, DNA damage checkpoint, microtubule-based process, and regulation of apoptotic process. The main signal pathways involved were cell cycle, FoxO signaling pathway, apoptosis, and p53 signaling pathway.

### 2.4. PPI Network Construction and Hub Genes Screening for Compounds ***14*** and ***6***

To gain a better understanding of cross-regulation of DEGs, a PPI network was established using an online visualization system. PPI network analysis for compound **14** indicated that the top 15 genes ranked as hub were PLK1, TUBA1A, CDK1, AURKA, CDC20, PCNA, CCVA2, CCNB1, CDKN1A, EGR1, JUN, TP53, FOXO4, CTNNB1, and CDK2 (Figure 3 and Figure 4). For compound **6,** the top 15 genes were TUBB4B, CDK1, PLK1, MCM5, AURKA, AURKB, CDK6, CDK2, CCND1, STAT1, CDKN1A, EGR1, TP53, BRCA1, and SUMO1 (Figure 5 and Figure 6). From our results, it is important to note that both of our compounds are closely related to genes and proteins involved in the microtubule network of the cell, such as the TUB family of proteins.

### 2.5. Comparison of Gene Expression Changes Caused by Compounds ***6*** and ***14*** with Gene Expression Profiles Induced by Treatment of Cancer Cells with Known Drugs

The DEGs obtained from the HeLa and PC-3 cells treated, respectively, with compounds **6** and **14** were compared to the gene expression profiles available in the CMap database and Drugbank, where the same cell lines were treated with known drugs (Figure 7 and Figure 8). The results depicted indicate that our compounds are both related to anti-tubulin drugs that are already known and used in the treatment of different types of cancer diseases. For example, compound **14** caused downregulation of the TUBA4A, TUBB, and TUBA1A genes, which is also closely related to the function of other drugs, such as vincristine, vinblastine, and colchicine (Figure 7). Likewise, compound **6** behaves similarly to **14** and other commercially available anti-tubulin drugs (Figure 8).

In fact, comparative analysis indicated that compounds **6** and **14** caused gene expression changes similar to those caused by microtubule inhibitors, strongly suggesting that both compounds could act as novel microtubule inhibitors (Table 3).

### 2.6. qRT-PCR Validation of Hub Genes

To validate the data elaborated in silico, HeLa and PC-3 cells were treated with compounds **6** and **14** and subjected to reverse transcription and qRT-PCR analysis. We focused the analysis on tubulin genes and, according to qRT-PCR results, the hub genes confirmed the results produced by the edgeR system, and the expression levels of the selected genes in the HeLa and PC-3 cells were consistent with the results of bioinformatic analysis. Compared to the control group, the expression of TUBA1A, TUBA4A, TUBA4B, and TUBB was downregulated (Figure 9).

### 2.7. Molecular Docking Studies

In a previous study, Arnst et al. confirmed that the cavity in colchicine microtubule composed of the three α-helices of Ala12, Asn206, and Leu227 was the docking site of this drug, which was also a potential domain of drug targeting due to its special spatial configuration [15]. Docking results showed that compound **14** and compound **6** could bind this site (Figure 10). Compared with compound **6** (LibDockScore, 100.56), compound **14** exhibited a higher docking score (LibDockScore, 115.39). Additionally, compound **14** inserted the entire molecule into the cavity with 1,4-dihydroindeno [1,2-b]pyrrole as the insertion site (Figure 10A). On the other hand, compound **6** inserted the cavity by chlorophenyl so that 1,4-dihydroindeno [1,2-*b*]pyrrole did not play an important role in binding (Figure 10B). In compound **14**, nitrogen atoms in selenadiazole were more likely to produce hydrogen bonds during the interaction in which the electronegativity of selenium atoms was weaker than that of sulfur atoms. Moreover, hydrogen bonds were critical for conformation and binding activity in ligand–receptor interaction. The interaction graph showed that nitrogen atoms in selenadiazole of molecule **14** formed hydrogen bonds with GLY143, producing π-sigma bonds with ALA12 and π-π stacked bonds with TYR224 with 1,4-dihydroindeno [1,2-*b*]pyrrole as the insertion site in molecular conformation. In addition, chlorine atoms on the benzene ring formed a halogen bond with ASN101 (Figure 11A). Compound **6** forced molecules to form π-π stacked bonds with TYR224 by chlorophenyl insertion through hydrogen bonding between sulfur atoms of thiadiazole and ALA12 and halogen bonds formed between chlorine atoms on benzene rings and GLN11 and GLN15 (Figure 11B).

### 2.8. Compounds ***6*** and ***14*** Interact with the Tubulin Polymerization Process

To further confirm that compounds **6** and **14** interact with tubulin assembly, a tubulin polymerization assay was performed. Figure 12 shows how paclitaxel, **6,** and **14** affect the process of tubulin polymerization. In particular, the action of compound **6**, which shows a higher speed in the nucleation process than the already known paclitaxel, [16] is surprising. Compound **14** also shows remarkable speed in the nucleation process and earlier plateau phase attainment. These results confirmed that both compounds significantly increase the polymerization of tubulin compared with the control. Moreover, compared to paclitaxel at equimolar concentrations, compound **6** in particular could bind with higher affinity by increasing the rate of polymerization, resulting in stabilization in less time. Therefore, compounds **6** and **14** could interact with the same binding site as paclitaxel and be considered as MTAs.

## 3. Discussion

Cancer remains one of the greatest challenges in the world of research and it is still one of the leading causes of death worldwide [17]. Currently, the main strategies to fight cancer are chemotherapy, radiotherapy, and surgery. Chemotherapy and radiotherapy have strong side effects, which can cause secondary pathologies and worsen the clinical picture of patients. For these reasons, researchers are trying to develop new anticancer drugs able to target specifically tumor cells while sparing normal cells, eliminating unwanted toxicities [18]. Thanks to the capacity of organizing the cytoskeleton and mitotic spindle, microtubules are crucially required for cell survival [19]. Given the higher mitotic index of cancer compared with normal cells, tubulin is a very well-known molecular target in cancer therapy, leading to the development of tubulin inhibitors [8]. Recently, MTAs are emerging as strong anticancer drugs both for the capacity to suppress tubulin aggregation into microtubules, causing apoptosis, and to inhibit angiogenesis [11,14,20].

1,3,4--Oxadiazoles derivatives have been shown to possess cytostatic and cytotoxic activity against cancer cell lines [21]. Mechanistically, their anticancer activity has been linked to inhibition of the epidermal growth factor receptor (EGFR) [21,22], vascular endothelial growth factor receptor (VEGFR) [23], or telomerase [24]. In this new study, we investigated the effect on the cancer cell cycle and gene expression of two 1,3,4 oxadiazole derivatives, compound **6** and **14** [1]. Flow cytometry experiments in HeLa and PC-3 cells showed that, after six hours from the beginning of treatments, both molecules are able to cause a decrease in cells residing in the G1 phase of the cell cycle and an increase in cells in S phase. A block in G2/M phase is observed after 18 h in HeLa and 12 h in PC-3 cells (Figure 1). There is a slight difference between the two cell lines in terms of cell cycle effects. HeLa respond better to treatment than PC-3, with a sharper peak of cells in G2/M. This could be explained by the fact that HeLa possess a faster and higher growth rate than PC-3. In fact, HeLa cells are known for their rapid proliferation and fast cell cycle, whereas PC-3 cells, although cancer cells, may exhibit some cell cycle alterations related to their metastatic nature. Drugs that modify the structure of microtubules destabilize normal cell functions and cell division, causing programed cell death and/or cell cycle arrest [25,26]. To understand whether the effects of compounds **14** and **6** were accompanied by changes in gene expression, HeLa and PC-3 cells were exposed for 18 h to a low concentration of the molecules (1µM) and subjected to RNA sequencing analysis. The results indicated that there were hundreds of genes activated or repressed in both HeLa and PC-3 cell lines. Notably, the number of DEGs appears to be slightly lower in PC-3 lines than in HeLa lines, results that are in line with what was obtained in the flow cytometry experiment in which the effect on the cell cycle appeared to be less pronounced in HeLa, especially regarding compound **6**.

GO and KEGG analyses showed that the compounds induced gene changes associated with microtubule motor activity and binding, as well as cyclin-dependent protein serine/threonine kinase activity (Figure 2). KEGG pathway analysis showed that a large fraction of DEGs were involved in DNA damage checkpoint and in the activation of cell death mechanism, including p53 signaling.

PPI network analysis demonstrated that compound **14** affected TUBA1A, CDK1, and TP53 expression (Figure 3 and Figure 4). On the other hand, compound **6** modified TUBB4B, STAT1, and BRCA1 expression (Figure 5 and Figure 6). All these genes are involved in crucial biological processes. Some mutants of TUBA1A were unable to assemble properly into a microtubule polymer, thus resulting in tubulin deficiency involving cell cycle abnormalities. Such abnormalities may contribute to the development of tubulinopathies, neurodevelopmental disorders, and carcinogenesis [27]. For these reasons, TUBA1A is a therapeutic target of some tubulin inhibitors already available, such as Taxol, nocodazole, and vincristine. TUBB4B, another member of the TUBA1A family, is a prognostic marker in endometrial, liver, and thyroid cancers. It has been shown that the downregulation of TUBB4B is crucial for the initiation of epithelial–mesenchymal transition (EMT), which is necessary for metastasis, although TUBB4B expression is upregulated in metastatic colorectal cancer lesions identified in sentinel lymph nodes [28]. Furthermore, it has been observed that, in colon cancer cells, TUBB4B influences cell polarity and controls focal adhesions [28]. Another gene affected by compound **14** was TP53; mutations on this gene affect activation of cell proliferation, suppression of DNA repair, and apoptosis. Therefore, dysregulations of TP53 levels lead to accelerated cell progression in several types of cancer, including prostate cancer and cervical cancer [29,30]. The CDK1 protein altered by compound **14** also has an important active role in cell cycle progression [31]. Much research has shown that dysregulation of CDK1 results in more aggressive tumor development, increased cell proliferation, and chromosomal instability. Compared with other kinases, altered CDK1 levels have more significant effects on cell progression. In addition, CDK1 has been found in several tumor types, such as ovarian carcinoma, liver carcinoma, breast carcinoma, colorectal carcinoma, and prostate cancer. Regarding BRCA1 affected by compound **6**, it has been seen that BRCA1 and BRCA2 pathogenic variants are associated with prostate cancer risk [32]. The BRCA1 gene provides instructions to produce a protein that acts as a tumor suppressor. Tumor suppressor proteins help prevent cells from growing and dividing too rapidly or uncontrollably. BRCA1 is involved in repair processes of damaged DNA by contributing to the stability of genetic information in cells. In addition, BRCA1 also regulates the expression of other tumor suppressors, proteins that regulate cell division, and plays an important role in embryonic development. Another gene affected by compound **6** was STAT1, which, in different tumor types (including prostate cancer), acts as a tumor suppressor and oncogene. Patients with localized prostate cancer who have high STAT1 expression have longer cancer-specific survival times, whereas patients with advanced prostate cancer who have low STAT1 expression experience early biochemical recurrence [33]. In addition, STAT1 abnormalities have been associated with increased resistance to docetaxel therapy against prostate cancer. Thus, it is evident that compounds **6** and **14** were capable of altering some of the most important tumor cellular processes and, in particular, cell proliferation. Therefore, based on the affected genes and the previous discussion, we evaluated the interactions of compounds **6** and **14** with known drugs. Both compounds showed similarities with drugs acting as tubulin inhibitors (Table 3). For instance, we found a significant similarity of compound **14** with fenbendazole (FZ). A recent study showed that FZ exhibits moderate microtubule depolymerizing activity against human cancer cells but possesses a potent antitumor effect in vivo and in vitro [34]. Furthermore, compound **6** showed a similarity of action to mebendazole. The latter has been evaluated in recent studies as a clinical candidate for use in the treatment of prostate cancer in synergy with docetaxel [35]. The binding of compounds **6** and **14** to tubulin and their mechanism of action were studied by deducing their crystal structure using RCSB PDB [15]. Both compounds showed binding specificity for a specific site in tubulin (Figure 10); however, molecule **14** exhibited a higher docking score for this site compared to molecule **6**. This could be explained by the fact that, differently from compound **6**, molecule **14** acts inserting the entire structure in the cavity with 1,4-dihydroindeno [1,2-*b*]pyrrole (Figure 10A). Compound **6** inserts the cavity by chlorophenyl and 1,4-dihydroindeno [1,2-*b*]pyrrole does not exhibit a pivotal role in the binding (Figure 10B). In agreement with these observations, both compounds were able to interfere with the tubulin polymerization process. Tubulin is a key component in tumor progression and both of our compounds were able to destabilize it to an even greater degree to paclitaxel at equimolar concentrations. Although the compounds appear to bind with higher affinity to the colchicine binding site, they behave as MSAs and mimic the action of paclitaxel. The observation that compounds **6** and **14** bind to the colchicine site on the tubulin beta chain but stabilize microtubules in a paclitaxel-like manner presents an interesting paradox. To analyze this phenomenon, it is necessary to consider the structural and functional implications of their binding, as well as the potential conformational changes induced by the compounds. The colchicine site is traditionally associated with destabilization of microtubule assembly. Colchicine itself, for example, binds to this site and prevents the addition of tubulin subunits, leading to microtubule depolymerization [36,37]. In the case of compounds **6** and **14**, while binding to the colchicine site, they show a stabilizing effect on microtubules. This suggests that their binding induces a different set of effects than colchicine. Indeed, binding of compounds **6** and **14** to the colchicine site may induce conformational changes in the structure of the surrounding tubulin, affecting the overall stability of the microtubule assembly. These compounds could potentially enhance tubulin–tubulin interactions within the microtubule reticulum, leading to increased stability rather than destabilization. It is important to consider that compounds **6** and **14** bind to the colchicine site differently from colchicine itself by only partially sharing the same amino acids. They interact with tubulin through unique molecular interactions and could induce specific structural rearrangements that promote microtubule stabilization. The specific binding modes of these compounds within the colchicine site may have a significant impact on their functional results. The above would demonstrate the similarity of our compounds in colchicine binding and tubulin stabilization function with paclitaxel, which is also supported by immunofluorescence data showing malformation of the microtubule complex during mitosis (Appendix A). Thus, the binding of compounds **6** and **14** to the colchicine site and the paclitaxel-like stabilization of microtubules suggest unique interactions and conformational changes that differ from traditional destabilizers, such as colchicine. Further investigation into their binding modes and structural effects on tubulin will be invaluable in unraveling the exact molecular mechanism underlying their stabilizing behavior. Despite these differences, perturbation of the tubulin–microtubule dynamic equilibrium leads to the same final result: metaphase arrest of cell division and induction of apoptosis.

In conclusion, this research showed that compound **6** and **14** target tubulin, perturbating normal cell cycle progression and causing the downregulation of genes involved in a spectrum of key biological processes. In the near future, we will conduct in vivo experiments to evaluate the efficacy of compounds **6** and **14** in reducing or arresting tumor growth and systemic effects in mouse models. This could initiate preclinical and clinical trials to compare their anticancer activity to current chemotherapeutic drugs targeting microtubules.

## 4. Materials and Methods

### 4.1. Cell Culture

HeLa (cervical carcinoma) and PC-3 (prostate adenocarcinoma) cell lines (ATCC, Rockville, Gaithersburg, MD, USA) were cultured in Dulbecco’s modified Eagle’s medium (DMEM) (Gibco, Grand Island, NY, USA). DMEM was supplemented with 10% of fetal bovine serum (FBS), 100 µg/mL streptomycin (Gibco), 1% of L-glutamine, and 100 units/mL of penicillin. Then, cells were incubated at 37 °C with 5% of CO_2_ in a humified incubator.

### 4.2. Cell Cycle Analysis

In order to understand the effect of compound **6** and compound **14** on cancer cells, we performed a cell cycle analysis. Both tumor cell lines were seeded on 6 cm dishes so as to achieve 50% confluence. After 24 h, cells were treated at a 1 µM concentration of compound 6 and compound 14 at different time points. After that, supernatant was aspired and discarded and cells were detached using trypsin and collected after having been centrifuged at 3000 rpm for five minutes. Then, the pellet was washed with cold PBS. Consequently, cells were fixed with 70% of cold ethanol on a vortex so as to break apart a cluster of cells and incubated o/n at −20 °C. Later, the pellet was resuspended in 0.4 mL propidium iodide staining solution (4ABiotech, Co., Ltd, Beijing, China) and incubated for 30 min at RT. Then, samples were acquired using BD FACS CANTO II and the next analysis was performed using Modfit LT^TM^ software.

### 4.3. RNA-Seq Library Construction and Sequencing

HeLa and PC-3 cells in the exponential phase of growth were plated into a 6-well culture plate according to the phenotype and population doubling of every cell line. Then, molecules **14** and **6** were added at a final concentration of 1 μM and cells were incubated from 6 to 18 h. After that, cells were collected at different time points. The RNeasy MINI Kit (Qiagen, Shanghai, China) was used for extraction of total RNA from cells. The Agilent 2100 Bioanalyzer system (Agilent, Co., Ltd., Shanghai, China) was employed to evaluate the quality of the extracted total RNA. Oligo (dT) magnetic beads were used for the isolation of poly(A) tail-containing mRNAs. Following that, the mRNAs were cut off randomly to establish a sequencing library with the mRNA fragments. After purification with the magnetic beads, 150–200 bp fragments were obtained to construct a PE150 library and perform sequencing with the HiSeq 500 Sequencer. Library construction and sequencing were undertaken by Novogene Co., Ltd., Beijing, China.

### 4.4. Identification of Differentially Expressed Genes

All data generated from the sequencer were converted to raw reads. In order to ensure the quality and reliability of data analysis, clean reads were obtained by removing the raw reads that contained adapters or unknown bases or had low quality. The clean reads were compared to the reference sequences before calculating the expression levels of genes and transcripts via RSEM. Intergroup differential expression analysis was conducted using edgeR.

### 4.5. GO Functional Annotation Classification and KEGG Enrichment Analysis

Key biological pathways associated with the DEGs were discovered via functional enrichment analysis. The Gene Ontology (GO) database was used for categorizing the selected DEGs by the involved biological processes, cellular components, and molecular functions. Kyoto Encyclopedia of Genes and Genomes (KEGG) was applied in the signaling pathway enrichment, annotation, and analysis of DEGs. GO functional enrichment analysis and KEGG pathway enrichment analysis were both carried out by online tools.

### 4.6. Protein–Protein Interaction (PPI) Networks and Hub Gene Analysis

To further investigate the mechanisms of the antitumor activity of 14 and 6 on the molecular level, a protein–protein interaction (PPI) network was set up using the STRING database, while an interaction network of DEGs was established based on the results of differential expression analysis and the interaction pairs available in the STRING database. In addition, the CytoHubba plugin of Cytoscape was used for the screening of central nodes in the network, and the genes with the top 15 Matthews correlation coefficient (MCC) values were considered as hub genes.

### 4.7. Connectivity Map and Drugbank Analysis of DEGs

The Connectivity Map (CMap) is a database of chemical interventions developed by the Broad Institute. The CMap database contains 7000 expression profiles that involve over 1300 compounds and it can be used for searching information of medications having similar effects to reveal the functional connections among SMCs, genes, and the morbid state. To recognize possible targets and the mechanism of action, highly correlated small-molecule drugs can be identified through comparative analysis by searching the CMap database for the upregulated and downregulated differential genes and the detection of similar small molecules with DrugBank database.

### 4.8. qRT-PCR Validation of DEGs

To verify the RNA-seq results, eight DEGs, identified by the sequencer, were isolated from different signaling pathways to form a qRT-PCR primer. Before the qRT-PCR, reverse transcription was performed to synthesize complementary DNAs (cDNAs) using the isolated RNAs. GAPDH was used as the reference gene and the relative expression level of every gene was determined using the 2^∆∆CT^ method.

### 4.9. Molecular Docking Verification

AutoDock is a suite of molecular docking tools designed to predict interactions between ligands and bio-macromolecular targets. It allows changes in conformation of small molecules and it evaluates molecular docking based on free energy changes. In this study, the microtubule proteins having a crystal structure were sourced from the PDB data downloaded from RCSB (extraction code: 5H7O). Ligands and water molecules were removed from the proteins, while 14 and 6 were used for the verification of molecular docking between the ligands and the microtubule proteins via AutoDock4.2.6. The docking results were subjected to PyMOL analysis.

### 4.10. Tubulin Polymerization Assay

The tubulin polymerization assay was performed using the kit provided by Cytoskeleton (BK006P, Cytoskeleton, Denver, CO, USA). Absorbance was recorded at 340 nm at 37 °C for 60 min with fixed acquisitions every 30 s. The assay was performed by incubating equimolar concentrations of tubulin with paclitaxel, compound 6, and compound 14. Tubulin without molecule was used as a negative control. After 60 min, the optical density of tubulin polymerization was recorded by reading the absorbance with the spectrophotometer. The experiments were performed in triplicate (Appendix A).

## Figures and Tables

**Figure 1 ijms-24-11263-f001:**
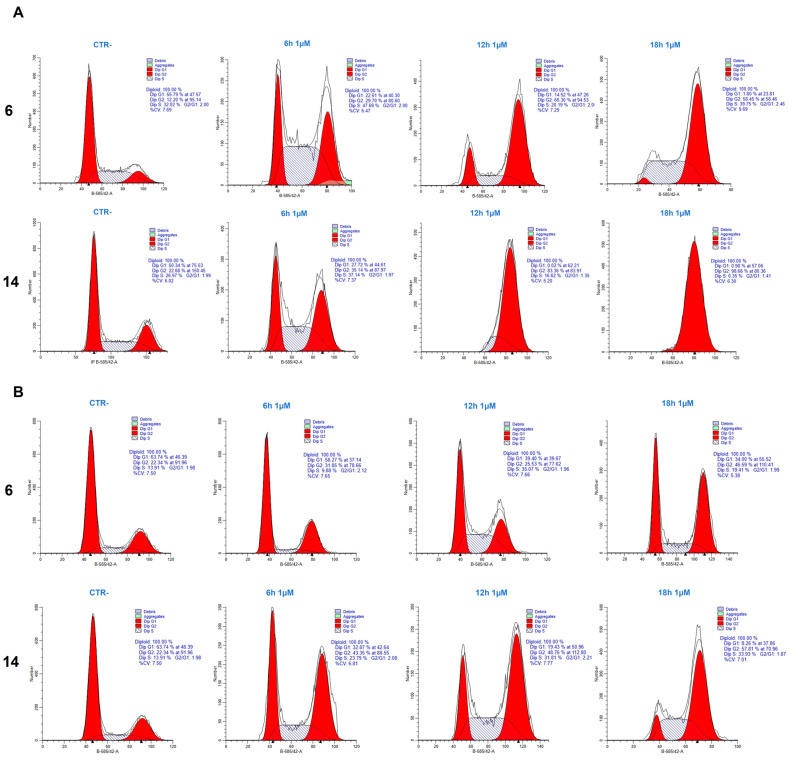
(**A**)**.** Cell cycle analysis of Hela treated with **6** and **14** at different time points. (**B**)**.** Cell cycle analysis of PC-3 cells after the administration of **6** and **14** at different time points.

**Figure 2 ijms-24-11263-f002:**
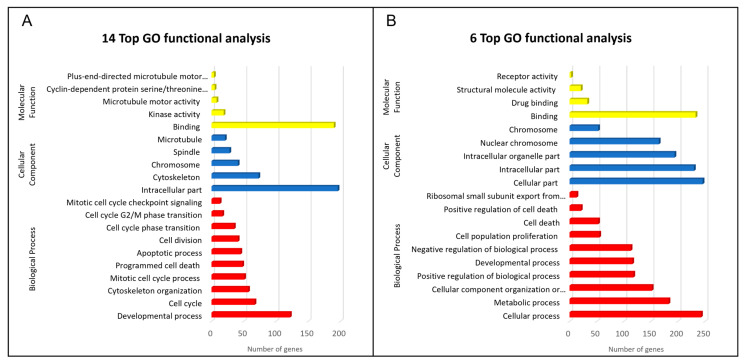
(**A**). Top GO functional analysis of DEGs of molecule **14**. (**B**)**.** Top GO functional analysis of DEGs of compound **6**.

**Figure 3 ijms-24-11263-f003:**
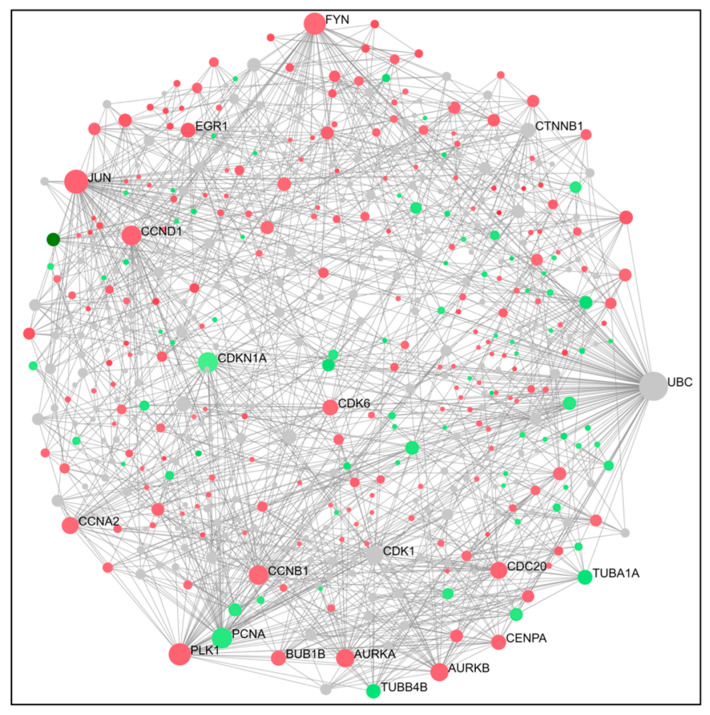
The PPI interaction network of DEGs (**14**). Concerning the figure, the nodes represent the differential genes and the connections between the nodes represent the interactions between the genes. Instead, the size of the nodes represents the degree of the genes, while red represents the up-regulation and green represents the downregulation.

**Figure 4 ijms-24-11263-f004:**
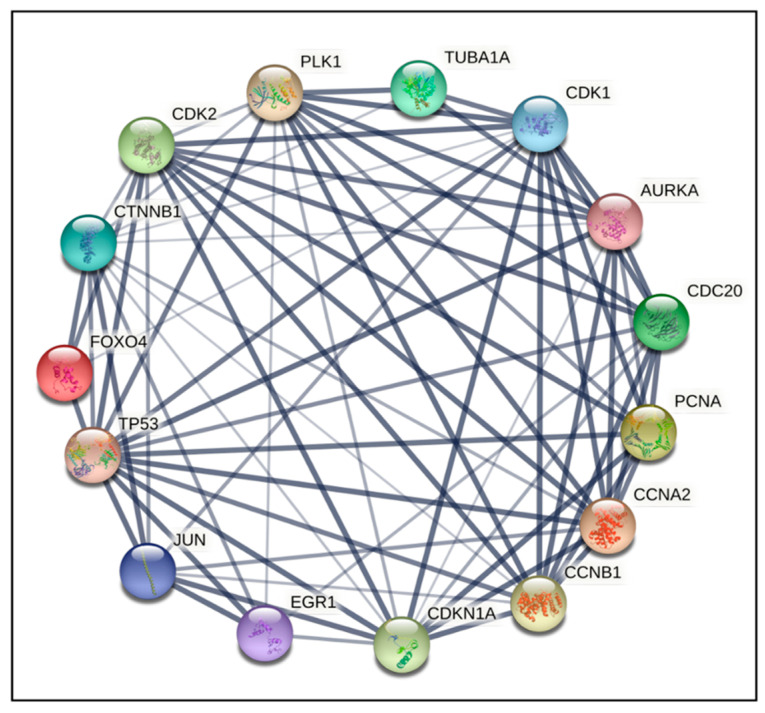
PPI network of the top 15 hub genes in **14**; network nodes represent proteins and edges represent PPI.

**Figure 5 ijms-24-11263-f005:**
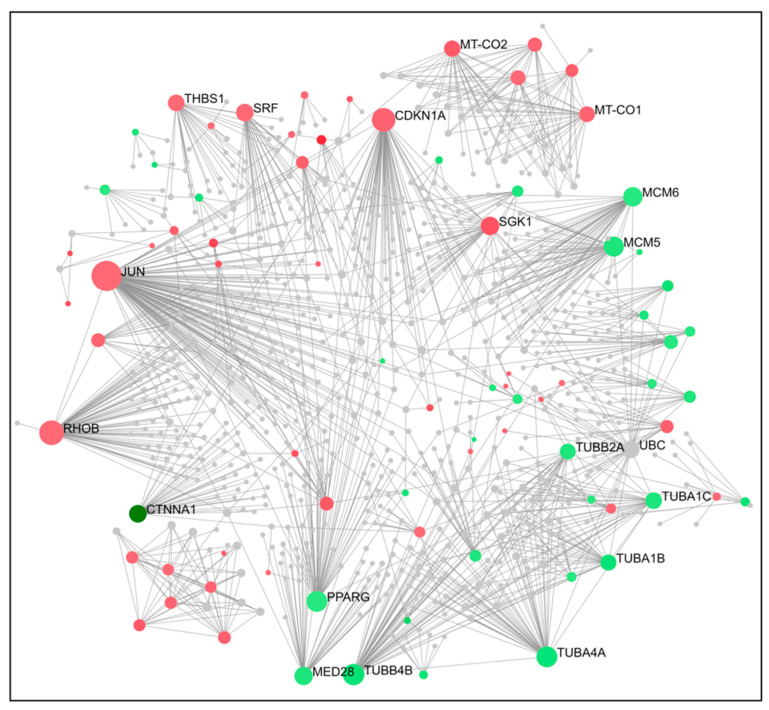
The PPI interaction network of DEGs (**6**); the nodes represent the differential genes, the connections between the nodes represent the interactions between the genes, and the size of the nodes represents the degree of the genes; red represents the upregulation and green represents the downregulation.

**Figure 6 ijms-24-11263-f006:**
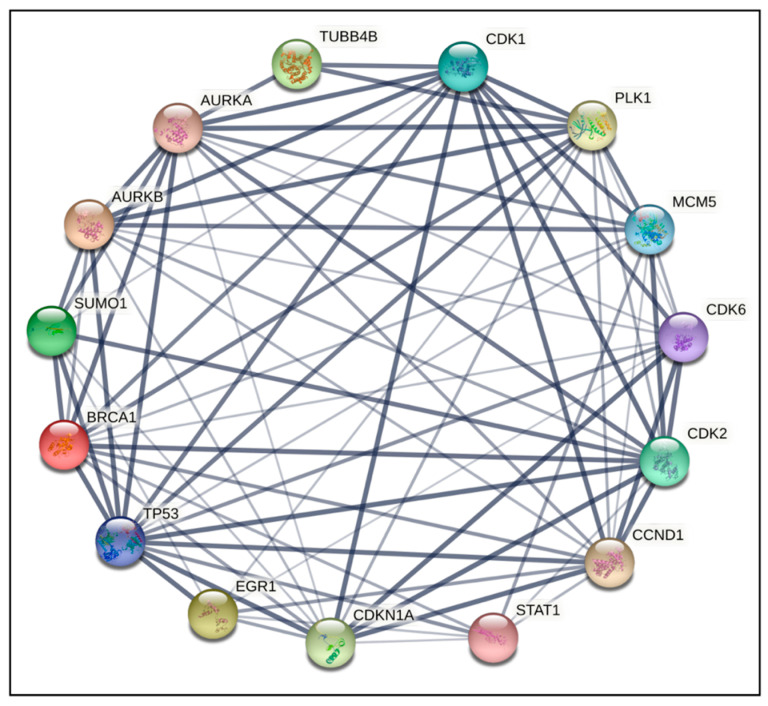
PPI network of the top 15 hub genes (**6**); network nodes represent proteins and edges represent PPI.

**Figure 7 ijms-24-11263-f007:**
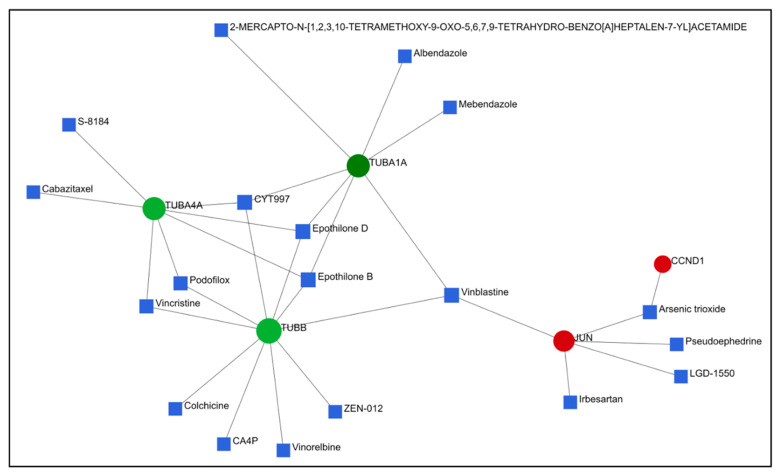
Protein–drug interactions of compound **14**. Downregulated genes are shown in green, upregulated genes in red, and known drugs that interact with them in blue.

**Figure 8 ijms-24-11263-f008:**
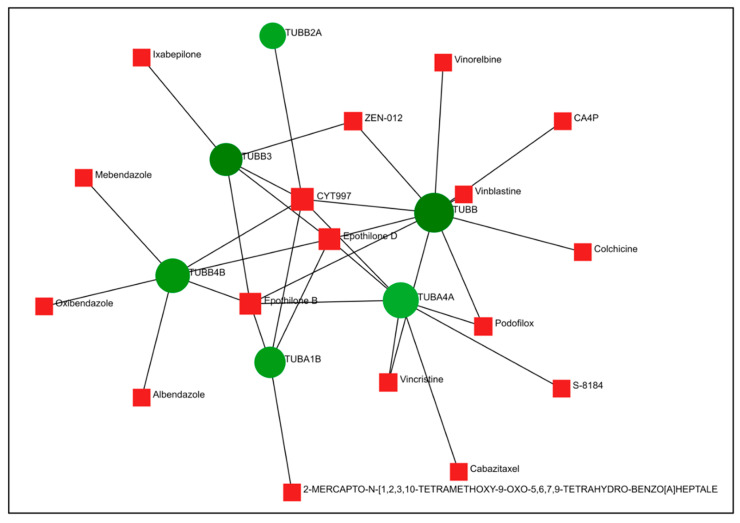
Protein–drug interactions of compound **6**. Downregulated genes are shown in green and known drugs that interact with them in red.

**Figure 9 ijms-24-11263-f009:**
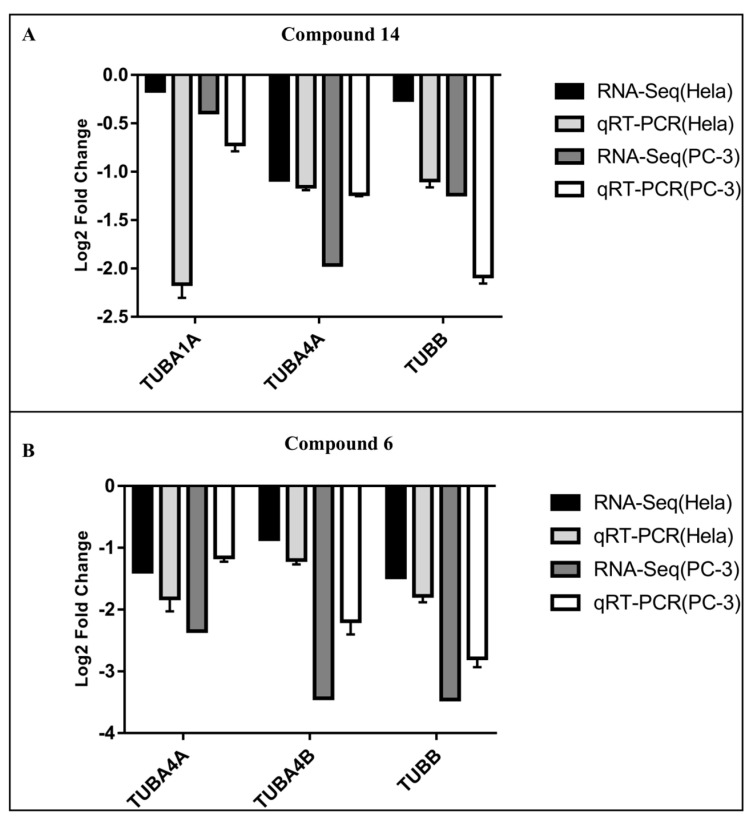
(**A**). Validation of the **14** RNA-Seq results by qRT-PCR. The x-axis shows the names of selected genes, while the y-axis represents the log2 fold change of gene expression. (**B**)**.** Validation of the **6** RNA-Seq result by qRT-PCR. The x-axis shows the names of selected genes, while the y-axis represents the log2 fold change of gene expression.

**Figure 10 ijms-24-11263-f010:**
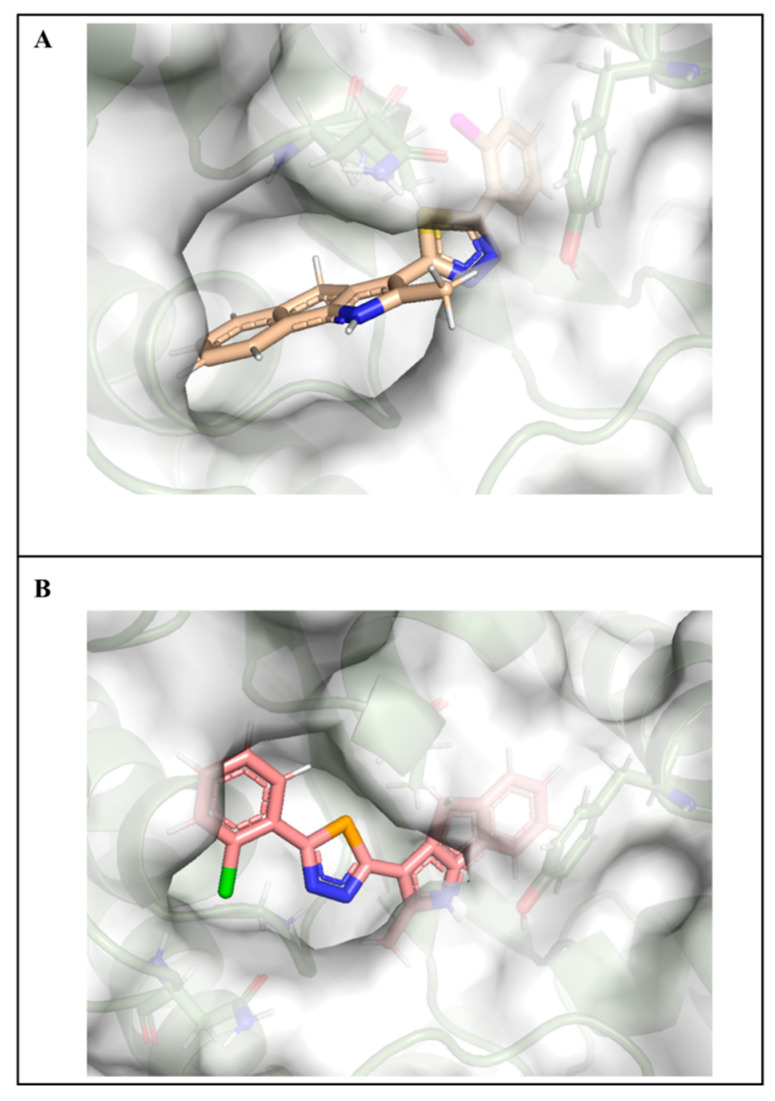
(**A**). Compound **6** surface diagram. (**B**). Compound **14** surface diagram. The colors represent different functional groups.

**Figure 11 ijms-24-11263-f011:**
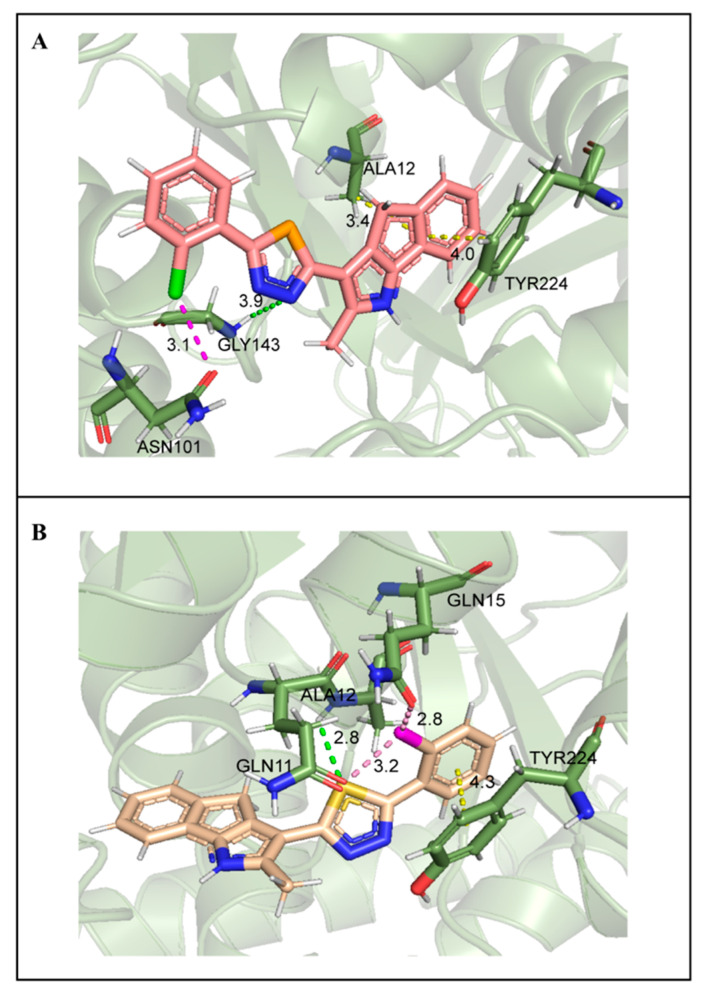
(**A**). Compound **14** interaction diagram. Dotted line: π-π stacked (yellow); halogen bond (pink); hydrogen bond (green). (**B**). Compound **6** interaction diagram. Dotted line: π-π stacked (yellow); halogen bond (pink); hydrogen bond (green).

**Figure 12 ijms-24-11263-f012:**
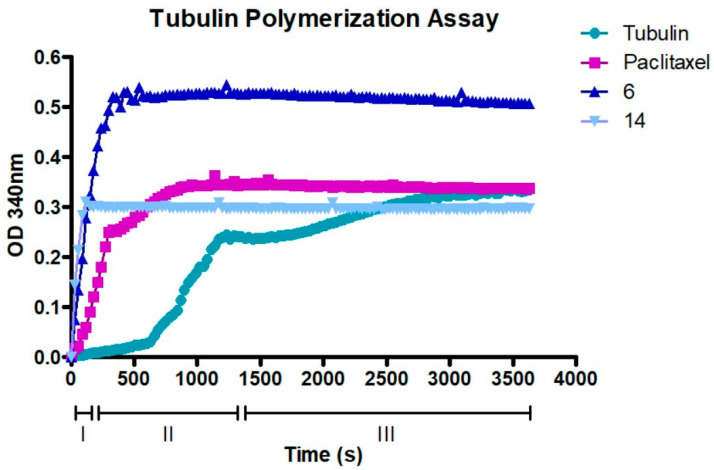
Tubulin polymerization assay. A standard polymerization curve (tubulin curve) is shown in the figure as a negative control. The three phases of polymerization of tubulin: I (nucleation), II (growth), III (steady state). Treated tubulin was preincubated with an equimolar concentration of paclitaxel, **6**, and **14** (10 µM). Then, tubulin polymerization was assessed by measuring absorbance every 30 s for a total of 60 min.

**Table 1 ijms-24-11263-t001:**
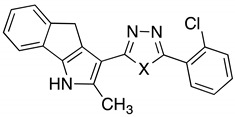
1,3,4-Thiadiazoles-2,5-disubstituted (**6**) and 1,3,4-selenadiazole analogues (**14**).

Compound	X	Compound	X
6	S	14	Se

**Table 2 ijms-24-11263-t002:** IC_50_ calculation of 1,3,4-Thiadiazoles-2,5-disubstituted (**6**) and 1,3,4-selenadiazole analogues (**14**) on HeLa and PC-3 cell lines.

Compound	X	IC_50_ HeLa (µM)	IC_50_ PC-3 (µM)
6	S	0.145 ± 0.06	0.176 ± 0.08
14	Se	0.005 ± 0.03	0.067 ± 0.06

**Table 3 ijms-24-11263-t003:** Pharmacologic perturbagens connected with molecules **6**- and **14**-induced DEGs.

	Name	Belongs to	Score
**14**	fenbendazole	Tubulin inhibitor	99.81
	oxibendazole	Tubulin inhibitor	99.81
	flubendazole	Tubulin inhibitor	99.8
	mebendazole	Tubulin inhibitor	99.78
	vincristine	Tubulin inhibitor	99.68
**6**	mebendazole	Tubulin inhibitor	99.87
	vindesine	Tubulin inhibitor	99.87
	fenbendazole	Tubulin inhibitor	99.82
	oxibendazole	Tubulin inhibitor	99.81
	flubendazole	Tubulin inhibitor	99.8

## Data Availability

The data presented in this study are available on request from the corresponding author. The data are not publicly available due to privacy.

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
