# Peer review of "RNAseq Analysis of Novel 1,3,4-Oxadiazole Chalcogen Analogues Reveals Anti-Tubulin Properties on Cancer Cell Lines"

_ijms, 2023, doi:10.3390/ijms241411263_

Round 1

Reviewer 1 Report

Reviewer Comments to Author:

Overall concerns:

The manuscript presents the antiproliferative activity of 1,3,4-oxadiazole derivatives against human cancer cell lines, specifically HeLa and PC3. These derivatives inhibit microtubule polymerization, leading to cell cycle arrest at the G2M phase. However, it is important to note that the findings regarding tubulin inhibitors have already been reported in previous studies by the authors, making them non-novel. Additionally, the use of tubulin and microtubule inhibitors is limited due to their side effects, such as neurological and myeloid toxicity. It would be valuable to highlight the advantages of these derivatives compared to other microtubule inhibitors. The authors need to clarify some points before recommending this manuscript for publication.

Specific Comments:

1. It is recommended to consolidate the scattered figures into a single, cohesive presentation or focus on the most important figures. Alternatively, select and present only the key figures that are crucial for understanding the manuscript.

2. Considering the comprehensive analysis conducted in this study, the title should be revised to accurately reflect the main claim of the manuscript. Additionally, the title should be revised to better align with the intended focus of the manuscript.

3. It is important to clarify whether the results obtained from HeLa and PC3 cell lines are similar or different. Provide a discussion regarding any observed variations between these cell lines.

4. In Figure 2, the histogram showing the number of differential genes is not useful for this manuscript. It is recommended to remove this figure or move it to the supplementary figure section if necessary.

5. In Figure 3, the gene ontology annotation analysis is interesting. However, the letter size is too small to understand. Please describe the pathway examination related to biological processes, molecular functions, and cellular components in the figure or provide a clear explanation in the text.

6. Regarding the Protein-Protein Interactions (PPI) and Protein-Drug Interactions analysis, the authors should present the results and discuss their significance in the figure or provide a thorough explanation in the discussion. If possible, authors should examine the effect of compound 6 and 14 on cancer cell lines carrying pathogenic variants.

7. In Figure 10, it is recommended to include a heat map of the RNA-seq data and qRT-PCR data to enhance the visual representation of the results.

9. In Figure 13, the tubulin polymerization assay results appear to have an irregular graph shape for the tubulin-only data. It would be beneficial to include results for colchicine and nocodazole in addition to taxol. Furthermore, explain any reasons for the differences observed in the plateau values for Compound 6, such as the possibility of variations in the amount of tubulin used. It is also recommended to include experiments with lower concentrations of Compound 6.

Please address these concerns and make the necessary revisions in your manuscript.

Reviewer 2 Report

The article titled Novel 1,3,4-Oxadiazole chalcogen analogues exhibit antiproliferative traits acting as microtubule inhibitors on cancer cell lines is accepted after consideration of the following comments.

1)     Abstract, it is not recommended to use we , us, I etc.

2)     I confused this new compounds or reported and if they are new authors should add spectral data and synthetic pathways.

3)     In the structure of compounds 6 and 14, 2 Cl is fixed in two structures so no need to mention as R.

4)     The study depends on only two compounds 6 1nd 14 so no rational and SAR discussion.

5)     Authors did not mention results of standard.

6)     Conclusion should improve

Round 2

Reviewer 1 Report

Reviewer Comments to Author: 

Overall concerns:

The revised manuscript has addressed the reviewer's comments. However, the authors need to improve the overall structure of the manuscript. The anti-mitotic activities of 1,3,4-oxadiazole derivatives against human cancer cell lines have already been reported in previous studies. Are these compounds novel? The authors should clearly state whether the compound is a tubulin stabilizing drug or a destabilizing drug and address some clarifications before recommending this manuscript for publication.

Specific Comments: 

The tubulin polymerization assay is not easy to obtain accurate results as tubulin polymerization is temperature and DMSO/glycerol-dependent. The authors mentioned that the number of tubulin polymerization assays was triplicate in the Materials and Methods section. It is recommended to provide additional control results, possibly in the form of a supplementary figure, to further support the findings.

The authors have cited a reference in the abstract. In MDPI journal, references are not typically cited in the abstract. Please remove the citation.

The authors mentioned that the binding site of the 1,3,4-oxadiazole chalcogen analogues to tubulin is the colchicine binding site. Colchicine, nocodazole, vinblastine, and vincristine bind to beta-tubulin, acting as tubulin polymerase inhibitors. On the other hand, compounds 6 and 14 are similar to Taxol in the in vitro polymerization assay. Taxol is one of the tubulin stabilizing drugs. Please explain the molecular mechanism by which compounds 6 and 14 stabilize the microtubules.

The authors have previously shown immunofluorescence images of HeLa cells treated with 0.5 µM 1,3,4-oxadiazole chalcogen analogues (compound 14). Generally, tubulin stabilizers such as taxol induce the formation of thick microtubule bundles during the interphase of the cell cycle in cultured cells. Additionally, they can lead to aberrant mitosis, characterized by the presence of multipolar spindles. Therefore, immunofluorescence should be performed in this study.

Tubulin does not undergo proliferation. Therefore, it would be more appropriate to refer to it as a tubulin polymerization assay instead of a tubulin proliferation assay.

Reviewer 2 Report

authors considered all comments and the article is accepted in the present form 

Author Response

We thank the reviewer. 

Round 3

Reviewer 1 Report

Reviewer Comments to Author:

Overall concerns: The revised manuscript has addressed the reviewer's comments. Please consider whether to include or remove the immunostaining data added in the supplementary figure.

Specific Comments:

The authors have included immunofluorescence images of HeLa cells treated with 0.5 µM 1,3,4-oxadiazole chalcogen analogues (compound 6 and 14) in the supplementary figure. It would be preferable to also include immunostaining data at 1 µM, similar to Figure 1. Additionally, the absence of a scale bar in each figure needs to be addressed. If these images are intended to be published, it is recommended to obtain permission from the journal or consider removing these figures from the manuscript. I was expecting the authors to provide further clarification on the molecular mechanism by which compounds 6 and 14 stabilize the microtubules.

I appreciate the opportunity to review your research. Thank you very much.
